# Revolutionizing colorectal cancer detection: A breakthrough in microbiome data analysis

Mwenge Mulenga[1,2]*, Arutchelvan Rajamanikam[3], Suresh Kumar[3]*, Saharuddin bin Muhammad[4], Subha Bhassu[4], Chandramathi Samudid[5], Aznul Qalid Md Sabri[6], Manjeevan Seera[7], Christopher Ifeanyi Eke[8]

1 Business Studies Division, National Institute of Public Administration, Lusaka, Zambia, 2 Centre for Research and Emerging Technologies, New Mulungushi, Kabwe, Zambia, 3 Department of Parasitology, Faculty of Medicine, University Malaya, Kuala Lumpur, Malaysia, 4 Institute of Biological Sciences, Faculty of Science, University Malaya, Kuala Lumpur, Malaysia, 5 Department of Medical Microbiology, Faculty of Medicine, University Malaya, Kuala Lumpur, Malaysia, 6 Faculty of Computer Science and Information Technology, University of Malaya, Kuala Lumpur, Malaysia, 7 School of Business, Monash University Malaysia, Selangor, Malaysia, 8 Department of Computer Science, Faculty of Computing, Federal University of Lafia, Lafia, Nasarawa State, Nigeria

* mwenge.research@gmail.com (MM); suresh@um.edu.my (SK)

## Abstract

The emergence of Next Generation Sequencing (NGS) technology has catalyzed a paradigm shift in clinical diagnostics and personalized medicine, enabling unprecedented access to high-throughput microbiome data. However, the inherent high dimensionality, noise, and variability of microbiome data present substantial obstacles to conventional statistical methods and machine learning techniques. Even the promising deep learning (DL) methods are not immune to these challenges. This paper introduces a novel feature engineering method that circumvents these limitations by amalgamating two feature sets derived from input data to generate a new dataset, which is then subjected to feature selection. This innovative approach markedly enhances the Area Under the Curve (AUC) performance of the Deep Neural Network (DNN) algorithm in colorectal cancer (CRC) detection using gut microbiome data, elevating it from 0.800 to 0.923. The proposed method constitutes a significant advancement in the field, providing a robust solution to the intricacies of microbiome data analysis and amplifying the potential of DL methods in disease detection.

## 1. Introduction

The emergence of Next Generation Sequencing (NGS) technology has ushered in a new era in clinical diagnosis and personalized medicine, enabling the analysis of human genome sequence-based data on an unprecedented scale [1]. NGS technology refers to high-throughput sequencing methods that enable rapid and cost-effective analysis of DNA or RNA sequences at a scale not previously attainable with traditional sequencing methods [2]. The availability of high-throughput data has enabled a wide range of applications in basic and applied research, clinical diagnosis, and personalized medicine [1]. NGS technology has made microbiome data accessible to the research community. Microbiome data, which refers to the

**Data availability statement:** The sequence data was deposited in the National Library of Medicine (NCBI) as a BioProject with accession number PRJNA1105667. The following is the URL to use: https://www.ncbi.nlm.nih.gov/bioproject/

**Funding:** Funding by Ministry of Higher Education (MOHE) Transdisciplinary Research Grant Scheme (TRGS/1/2018/UM/01/7).

**Competing interests:** The authors have declared that no competing interests exist.

genetic material of the microorganisms present in a particular environment or sample, has the potential to be used for clinical diagnosis and personalized medicine.

The human microbiome, which refers to the microorganisms that live in the human body, has been found to play an important role in human health and diseases such as colorectal cancer (CRC) [3]. Several studies have used microbiome data to predict CRC [4–6]. However, microbiome data has several limitations, including high dimensionality and noise, which make it difficult to analyze using traditional statistical methods. Microbiome data is often subject to variability due to factors such as sample collection, sequencing technology, and data processing. Machine learning (ML) algorithms, which perform comparatively better than traditional statistical methods on high-dimensional data, are more robust to noise and can effectively identify patterns in noisy data [7]. However, traditional ML methods also face limitations when applied to microbiome data, such as data sparsity and the need for hand-crafted feature selection methods [8]. Consequently, the research community has been leaning towards the use of deep learning (DL) in disease identification using microbiome data, due to its success in related fields such as computer vision, natural language processing, and image analysis [9].

Recently, deep learning (DL) has been extensively applied in the detection of colorectal cancer (CRC) using gut microbiome data. However, DL methods often face challenges due to the high dimensionality and noise present in microbiome data, which can degrade model performance. Several studies have implemented DL methods combined with feature extraction techniques, such as autoencoders [10] and unsupervised binning [11]. Other scholars have also used feature selection techniques. For instance, the works in [12,13] and [14] used filter-based feature selection in their microbiome data-based models for disease identification. Despite these advances, many existing methods take a piecemeal approach, often only partially addressing the issues of dimensionality and noise in DL-based disease detection. Baseline work in [15] proposed a "chaining of normalization" technique, which applies sequential normalization methods to the input data, but it does not fully address the critical challenges of dimensionality reduction and feature relevance.

This study proposes a novel hybrid method that integrates chained normalization, rank transformation, feature extension, and feature selection to enhance the performance of DL models in CRC classification based on microbiome data. Hybrid methods in predictive modeling combine the strengths of multiple techniques or algorithms to overcome the limitations of individual methods [16–18]. The proposed method also emphasizes data preprocessing, which reduces noise in microbiome data to enhance disease identification. Enhancing CRC classification accuracy is critical for early detection and effective intervention. Leveraging advanced computational methods to improve CRC classification accuracy holds immense potential for enhancing patient outcomes and reducing healthcare burdens. This approach addresses the limitations identified in existing studies and makes significant contributions to the field, as outlined below:

- Development of a hybrid method combining chained normalization, rank transformation, and feature selection. While chained normalization attempts to adjust the data distribution towards a normal distribution, and rank transformation attempts to reduce noise in the data, feature selection is used to tackle the dimensionality issue in microbiome data. The proposed hybrid method enhances model performance due to the cumulative effect of its constituent methods. Normally distributed data lead to more reliable results in statistical models, noise reduction leads to less bias, and dimensionality reduction reduces overfitting in a model.

- Integration of chained normalization, rank transformation, feature extension, and feature selection to enhance the quality of features while addressing the dimensionality and noise

issues in microbiome data. While the combined methods are meant to improve the quality of features in the dataset, feature extension generates additional features that may have more relevance to the model. These additional features may not necessarily lead to high dimensionality in the model, since feature selection is later applied as a countermeasure.

- Application of the proposed method to a new Malaysian dataset, demonstrating its practical utility.

The structure of this paper is as follows: Section 2 provides a comprehensive review of the relevant literature. Our unique methodology is detailed in Section 3. Section 4 is dedicated to the presentation and discussion of our results. The paper concludes with Section 5, where we summarize our findings and their implications.

## 2. Related works

This section discusses pertinent work addressing the challenge of noise in microbiome data. Various researchers have implemented techniques that transform data from continuous to categorical values. For instance, Nguyen and Zucker [11] proposed one-dimensional representations of microbiome data using unsupervised binning methods and scaling algorithms to enhance the predictive performance of disease detection in artificial neural networks. They explored various discretization methods based on the frequency, width, and presence of a feature in a data sample. In another study, Yazdani et al. [19] leveraged ML to identify shifts in gut microbiome abundance due to diseases. They discretized the probability outputs of a classifier into under-abundant, over-abundant, and "neutral" categories, which were then used to train another classifier on the extracted features. Similarly, Lustgarten et al. [20] proposed a Bayesian discretization approach to optimally categorize features of high-dimensional biomedical data. This method incorporates a Bayesian score component for evaluating discretization and a search procedure to efficiently identify possible categorizations. While discretization methods mitigate the effects of minor observation errors and eliminate data noise, they may also lead to a loss of information.

Feature generation has also been employed to enhance the performance of predictive models. For example, Katz, Shin, and Song [21] proposed a framework for automating feature generation, where a large set of candidate features was generated through the combination of variables in the original dataset. Their method adds flexibility to a predictive model by using criteria-based feature selection to further improve performance. However, their work was based on traditional ML methods, which may be prohibitive for non-domain experts. In a related work, Kaul, Maheshwary, and Pudi [22] proposed a feature generation approach that mines feature associations between variables using a regression method to identify stable relationships. Although the method does not require domain knowledge from the users, it is also based on traditional ML methods and may be limited by data sparsity in microbiome datasets. Similarly, Wang, Gu, and Wang [23] proposed an SVM-based framework that applies feature augmentation to effectively detect intrusions in computer networks. They implemented logarithmic marginal density ratios to transform original features into new ones, thereby improving the predictive performance of the model. Although their method outperformed existing approaches, it was not tested on microbiome data.

Feature selection plays a crucial role in predictive models. For instance, Ananthakrishnan et al. [24] utilized a knowledge-guided feature selection technique in an ANN-based model to predict responses to anti-integrin biologic therapy using a combination of microbiome and clinical IBD data. While the integrative feature selection used in the experiment may add interpretability to the model, manual feature selection can be challenging for non-domain experts. Similarly, integrative feature selection was employed by Dadkhah et al. [13], who

utilized four statistical tools namely, Kruskal-Wallis, LEfSe, MetaStats, and Indicator Species Analyses to identify informative operational taxonomic units (OTUs) in various polyp groups. Each statistical method identified a subset of relevant features, which were then used in separate models. Additionally, Pietrucci et al. [25] used embedded feature selection, subjecting a random forest (RF) algorithm to a subset of features that increased gradually while observing model performance to establish the optimum number of features. The subset of features that achieved performance close to the target was considered the most relevant. However, embedded feature selection methods are generally computationally intensive. In a related study, Sharma, Paterson, and Xu [26] clustered simulated OTU data and applied ensemble feature selection to improve accuracy and optimize computational performance. While ensemble feature selection methods offer outstanding performance, they are prone to computational complexity in predictive models.

Data normalization, a crucial step in the preprocessing stage of the ML pipeline, presents significant limitations in microbiome data [27]. Commonly accepted normalization methods, such as Median and Median Absolute Deviation Normalization (MMADN), Sigmoid Normalization, Z-Score Normalization (ZSN), Minimum-Maximum Normalization (MMN), Variable Stability Scaling (VSS), and Pareto Scaling Normalization (PSN), have shown limited performance in gene sequence data [15,28]. This is because microbiome count data is compositional, implying it represents a constant sum and non-negative values [29]. Normalization methods recommended for gene sequence-based data, such as Rarefying, Relative Log Expression (RLE), and Trimmed Mean of M-values (TMM), also fail to account for sparsity in the data [30,31]. Research findings suggest that no single normalization method is robust across all datasets, as the generalization capacity of individual methods is limited by the varying underlying properties of different datasets [32].

In light of the aforementioned drawbacks identified in existing studies, this paper proposes a method that combines chained normalization, rank transformation, feature extension, and feature selection to improve feature relevance and address the high dimensionality and noise associated with microbiome sequence data in DNN-based classification of CRC. The work in this paper is unique in that, while existing methods propose chaining and stacking of normalization techniques separately, this study integrates the two techniques as a hybrid approach. Additionally, a variation is introduced where chaining precedes ranking, while in previous works, data ranking preceded chaining. This study also employs these two variations to generate two sets of features, which are then merged as a mechanism for feature extension. Following the feature extension stage, feature selection is applied to enhance the predictive performance of the DNN model.

## 3. Methodology

This paper introduces a methodology that amalgamates chained normalization, rank transformation, feature extension, and feature selection to tackle the challenges of high dimensionality and noise inherent in microbiome sequence data for deep neural network (DNN)-based colorectal cancer (CRC) classification. The experiments were conducted using Python 3, leveraging the Keras framework [33] and TensorFlow as provided by Google Colab [34]. The microbiome data used in this study was collected from human subjects recruited for a study conducted at the University of Malaya Medical Centre.

### Ethics statement

The study design involving patients was approved by the University of Malaya Medical Centre (UMMC) Medical Research Ethics Committee (MRECID: 201914–6975). Patients and the

public were first involved during fecal sample collection and questionnaire administration. Recruited individuals were either identified by healthcare professionals or voluntarily admitted. All research questions and outcome measures were approved by the UMMC Medical Research Ethics Committee and were explained in detail to each participant by the enumerators and clinicians. There was no involvement of patients or the public in the design of this study. Participants verbally agreed to have their results published. All participants provided verbal and written consent to be part of the study.

## Dataset

Two datasets were used in this paper, including our own dataset collected from the University of Malaya Hospital and one additional dataset adopted from the work in [15]. In the remainder of this paper, the dataset from the University of Malaya Hospital will be referred to as Dataset 1, while the other dataset will be referred to as Dataset 2 and, as described below. Dataset 1 comprises 112 samples and 324 features, selected based on the validity of the disease status (CRC) entry. Records with missing values in the disease status field were excluded. The subsequent subsections detail the data collection and preprocessing into OTUs.

## Study population and fecal specimen collection

Fecal sample collection was carried out from June 2019 to January 2021 under the Gut Health Project for patients attending the Colorectal Surgery Clinic at the University of Malaya Medical Centre (UMMC), Malaysia. Patients were either referred by the primary clinic at UMMC or other private hospitals based on risk factors and red-flag symptoms. The control group consisted of individuals from colorectal screening campaigns and general stool screening surveys conducted around Klang Valley. From the UMMC patients, a total of 136 were approached, and 88 consented to participate in the study. Of these, 33 were diagnosed with colorectal cancer following clinical and pathological evaluations. The control group comprised non-cancer individuals who tested negative for the fecal occult blood test. The study population was primarily urban, with a mean age of 55, and included the three major ethnic groups: Malay, Chinese, and Indian. Stool samples were collected early in the morning using sterile stool containers provided to the participants. After collection, the samples were processed within three hours and stored at −80°C.

## DNA extraction and library preparation for 16S metagenomics

Fecal genomic DNA was extracted using the QIAamp PowerFecal DNA Kit. The extracted DNA was then used to amplify the 16S hypervariable region (V3–V4 region) of bacterial DNA. The resulting amplicons were verified and proceeded to library preparation following the workflow established by Illumina. Sequencing was performed on the Illumina MiSeq platform and the sequences were deposited in the National Library of Medicine (NCBI) as a BioProject with accession number PRJNA1105667.

## 16S metagenomic data analysis

Sequence analysis was conducted using widely adopted platforms and packages in Qiime2 and R software. The raw reads were initially preprocessed to retain only high-quality reads using algorithms such as DADA2. The amplicon sequence variants (ASVs), consisting of only high-quality reads, underwent phylogenetic assignment. The abundances of different taxonomic groups (at the genus, family, order, class, and phylum levels) in each study group were determined by normalizing the number of reads assigned to each group in the corresponding gut metagenome against the total number of sequenced reads in the entire metagenome.

## Additional dataset

The second dataset is constructed from meticulously selected samples of shotgun sequence data, with detailed information provided in reference [35]. This dataset comprises 884 samples and 2,031 features. Among these, there are 368 CRC samples and 516 controls. The dataset includes samples from six countries: the United States of America (USA), Canada, France, Germany, Austria, and Italy.

## Adopted normalization methods

Initially, six normalization methods, namely, Sigmoid, MMADM, MMN, PSN, VSS, and ZSN were considered for preliminary analysis in the experiment to select the most optimal methods. Preliminary results showed that Sigmoid and MMADN demonstrated superior results. Therefore, the proposed method employs four normalization methods: MMN, PSN, VSS, and ZSN, which are used as input during chained normalization. During this process, the normalization methods are sequentially invoked to transform the input data, also known as raw data. The sequential arrangement implies that each normalization method takes turns transforming the data, with the output of one method being fed into the subsequent normalization method. Research has shown that no single normalization method outperforms others across all datasets due to the varying nature of underlying data distributions [32]. Similarly, no optimum combination of normalization methods exists for use in a chained sequence for the same reason. Thus, our decision to apply multiple normalization techniques was motivated by the need to capture different aspects of the data that a single method might overlook. Previous work used brute force to identify an optimal combination of normalization methods in their proposed chained technique [15]. The four normalization methods utilized in this study are described in Table 1.

As can be observed in Table 1, while the MMN technique adjusts the data scale and preserves the relationships between the original and rescaled data, ZSN, VSS, and PSN are robust in mitigating the impact of dominant features and outliers in a dataset [32].

## The proposed hybrid method

Our proposed hybrid method combines chained normalization, rank transformation, feature extension, and feature selection techniques. The key objective is to create two additional sets of features from the raw data, which are subsequently merged. After merging, these features undergo feature selection before training the DNN model. Fig 1 presents a step-by-step diagram of the feature engineering process utilized in the experiment.

**Table 1. Description of normalisation methods used in the current work.**

| Formula | Description |
|---|---|
| $a'_{i,j} = \dfrac{a_{ij} - \mu_j}{\sigma_j}$ | ZSN method normalizes data features by setting their mean to zero and variance to one |
| $a'_{ij} = \dfrac{a_{ij} - \mu_j}{\sqrt{\sigma_j}}$ | In the PSN method, the scaling factor used is the square root of the variance of the data. |
| $a'_{i,j} = \dfrac{a_{ij} - \mu_j}{\sigma_j} * \dfrac{\mu_j}{\sigma_j}$ | VSS method aims to enhance the Zero-mean and Unit Variance Standardization (ZSN) by introducing a scaling factor called the coefficient of variation. |
| $a'_{ij} = \dfrac{a_{ij} - min(a_j)}{\max(a_j) - \min(a_j)}(iMax - iMin) + iMin$ | MMN is value-based normalization method which is useful for preserving the relationships among the original input. |

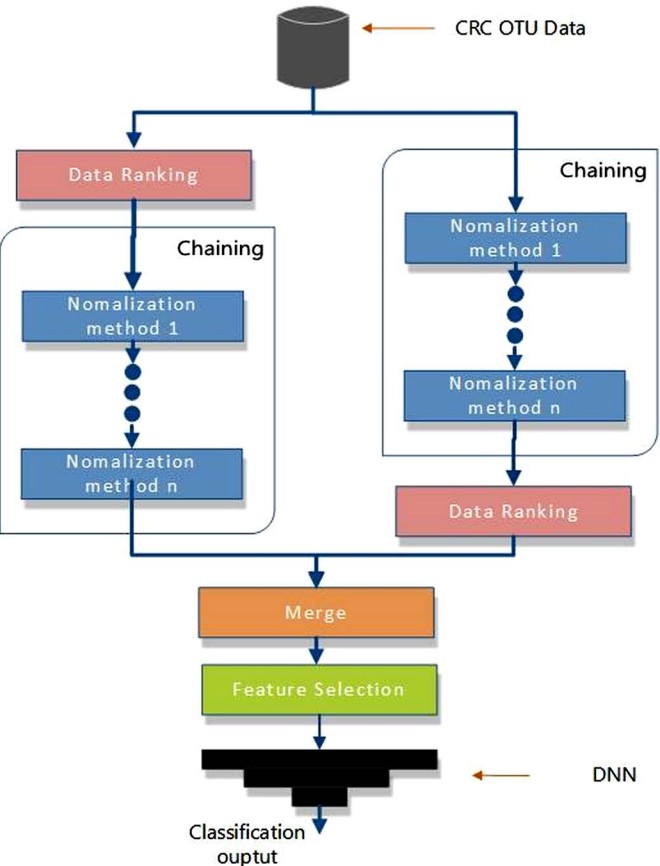

**Fig 1. Feature generation-based data normalization, ranking, and feature selection.**

As depicted in Fig 1, the algorithm operates along two parallel paths to generate new features from the raw input data. These features are then merged into a single dataset, which undergoes feature selection. Both execution paths involve the same processes, specifically chaining and rank transformation. However, the order of these processes differs between the two paths. In the first path, rank transformation is applied before chained normalization, while in the second path, the sequence is reversed. These two distinct routes produce two different datasets, and the entire algorithm results in a completely new dataset once the two intermediate datasets are merged. It is important to note, as discussed earlier, that due to the varying nature of underlying data distributions, it is not currently possible to determine the optimal sequence for arranging these processes in the algorithm shown in Fig 1. Therefore, several configurations were tested during the experiment to identify the best combination in terms of predictive performance. The idea of merging features in a dataset with newly generated features was also employed in previous research, yielding positive outcomes [15]. Additionally, feature selection is performed to enhance classification accuracy and reduce the execution time of the DNN model by addressing the high dimensionality inherent in microbiome data. The pseudo code for the algorithm is provided in Algorithm 1.

Algorithm 1 starts with loading normalization methods in a list called *nomalizationMethodsList*, which is used to generate a collection comprising all possible combinations of the normalisation methods. The list containing the combinations identified is assigned to a variable called *nomalizationMethodsCombinationsList*. The function then loops over this newly

**Algorithm 1. Feature engineering using normalization, ranking, feature generation and feature selection.**

**Input**: List of normalisation methods, raw dataset
**Output**: Five new dataset
1: Function generatDataSets
2: *normalizationMethodsList* ← [list of normalisation methods]
3: *normalizationMethodsCombinationsList* ← []
4:
5: # Get all possible combinations of normalisation methods
6: **for** r in range (1, len(normalisationMethodsList) + 1):
7: *combinations* ← itertools.combinations(normalizationMethodsList, r)
8: normalizationMethodsCombinationsList.extend(list(*combinations*))
9:
10: # Initialise variables
11: *chainedDataset* ← raw data
12: *preRankDataset* ← chainedDataset
13: *chainedPreRankDataset* ← []
14: *chainedPostRankDataset* ← []
15: *PreAndPostRankDataset* ← []
16:
17: # Normalise and rank chainedDataset
18: **while** *normalizationMethodsCombinationsList:*
19: *normalizationMethodsCombination* ← normalizationMethodsCombinationsList.pop()
20: **for** method in normalizationMethodsCombination:
21: *chainedDataset* ← method.normalise(*chainedDataset*)
22: chainedDataset ← columnwiseRankTransformation(*chainedDataset*)
23: chainedPostRankDataset.append(chainedDataset)
24:
25: # Normalise and rank preRankDataset
26: **for** *method* in *normalizationMethodsList:*
27: preRankDataset ← *method*.normalise(*preRankDataset*)
28: **for** *normalizationMethodsCombination* in *normalizationMethodsCombinationsList*:
29: **for** *method* in *normalizationMethodsCombination*:
30: *preRankDataset* ← method.normalize(*preRankDataset*)
31: chainedPreRankDataset.append(*preRankDataset*)
32: # Merge chainedPreRankDataset and chainedPostRankDataset horizontally
33: **for** *i* in range(len(*chainedPreRankDataset*)):
34: PreAndPostRankDataset.append(np.hstack((*chainedPreRankDataset[i]*, *chainedPostRankDataset[i]*)))
35: *normalizedDataSetFS* ← lasso(*PreAndPostRankDataset*)
26: train *model* on *normalizedDataSetFS*
37: *modelPerformance* ← evaluate *model* on *normalizedDataSetFS*
35:
38: return *modelPerformance*
39:
40: End Function
41:
32: End Function

https://doi.org/

generated list to retrieve a collection of normalization methods in combination identified in the previous step and assigns it to a variable called *nomalizationMethodsCombination*. Now, during the chaining of normalization methods step, the nested loop is used to obtain each of the normalization methods from *nomalisationMethodsCombination*, to transform the raw dataset one after the other. The normalized dataset is stored in the variable called *chained-Dataset*. Column-wise rank transformation is then applied to the normalized data and the result is assigned to the *chainedPostRankDatase*t variable.

To generate a contrasting copy of the intermediate dataset obtained in the previous steps, a copy of a raw dataset stored in the *preRankDataSet* variable is transformed using a loop. Rank transformation is performed on it and thereafter the output is subjected to chain-based

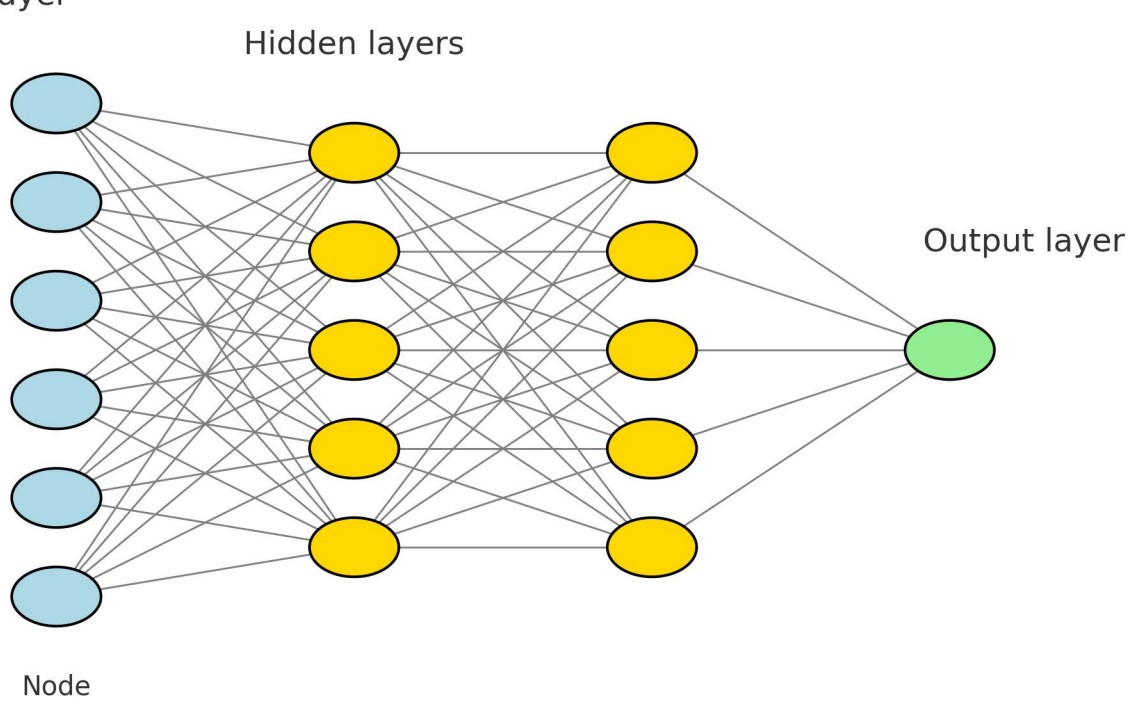

**Fig 2. An architecture of a deep neural network.**

normalization. The normalized data is stored in the variable called *chainedPreRankData-set*. Next, the two variations of the original dataset, namely, *chainedPreRankDataset* and *chainedPostRankDataset* merged along the second axis (horizontal merging) and the result was assigned to the variable called *preAndPostRankDataset*. Feature selection is performed on the merged data and the output is assigned to the variable called *normalizedDataSetFS*. The feature selection method used in this algorithm is based on L2 (Ridge) regularization adopted from the work in [36]. The threshold value is set 0.1 to determine which features are important. Features with importance scores above this threshold are selected for inclusion in the final feature set. Finally, the output of feature engineering is fed into the DNN algorithm to predict CRC. The python-based script of the algorithm outlined above can be found the following link: https://github.com/diskava/Revolutionizing-colorectal-cancer-detection-A-break-through-in-microbiome-data-analysis.git

## The DNN architecture

This research employs a 4-layer Deep Neural Network (DNN) architecture, based on modifications and fine-tuning of the model from [15]. The architecture consists of 512 nodes in the first hidden layer and 256 nodes in the second hidden layer. The model's input layer contains 324 neurons, and there is a single neuron in the output layer. The hidden layers utilize the

Rectified Linear Unit (ReLU) activation function, while the output layer employs a sigmoid activation function. The optimization algorithm used is Adaptive Moment Estimation (ADAM), with a learning rate set to 0.001, consistent with previous work. The DNN architecture is depicted in Fig 2.

## 4. Results and discussions

The previous section detailed the hybrid feature engineering method proposed in this research, which integrates chained normalization, rank transformation, feature extension, and feature selection to minimize noise and address dimensionality reduction in microbiome data. The aim of this methodology is to improve CRC classification performance using a DNN model. This section presents the results obtained through these methods.

The goal of the research is twofold: first, to demonstrate that the improved performance of the proposed model results from the combined strength of each constituent component; and second, to show that the DNN model's performance improves incrementally as the components are combined. To this end, the performance of the DNN model is monitored cumulatively as components are added gradually, and the intermidiate results serve as a baseline for comparison with the final target model.

### Performance metrics

To assess the efficacy of the proposed methodologies and standard approaches, three performance metrics are employed: sensitivity, specificity, and area under the curve (AUC). The calculation of these metrics relies on a two-by-two confusion matrix.

- The True Positive (TP) metric is found in the upper-left cell of the matrix, representing the count of accurately classified colorectal cancer (CRC) entries.

- The False Positive (FP) metric, located in the upper-right quadrant, indicates the count of CRC samples incorrectly classified as healthy.

- The False Negative (FN) metric is displayed in the lower-left corner, representing the number of healthy samples that were mistakenly classified as CRC.

- The True Negative (TN) value, found in the bottom-right cell, signifies the accurate classification of healthy samples.

Sensitivity, also known as the True Positive Rate (TPR), is defined as the ratio of correctly identified cases of CRC to the total number of actual CRC cases. It can be mathematically calculated as follows:

$$\text{Sensitivity} = \text{TPR} = \frac{TP}{TP + FN} \tag{1}$$

Specificity, which is also referred to as the True Negative Rate (TNR), represents the accuracy of correctly identifying healthy samples and is calculated as follows:

$$\text{Specificity} = \text{TNR} = \frac{TN}{TN + FP} \tag{2}$$

The false positive rate refers to the ratio of negative cases that are incorrectly classified as positive cases. This metric is calculated as follows:

$$\text{FPR} = \frac{FP}{TN + FP} \tag{3}$$

The Receiver Operating Characteristic (ROC) curve is a visual tool that illustrates the relationship between the False Positive Rate (FPR) and the True Positive Rate (TPR) by plotting them on the horizontal and vertical axes, respectively. The horizontal axis represents the False Positive Rate (FPR), which indicates the ratio of healthy samples incorrectly classified as colorectal cancer (CRC) samples. Conversely, the vertical axis denotes the True Positive Rate (TPR), corresponding to the ratio of accurately classified CRC samples. The Area Under the Curve (AUC) is a metric used to quantify the extent of the area beneath the ROC curve. A desirable model is characterized by high sensitivity, specificity, and AUC.

## Analysis of the model's performance

The results of the model and baseline methods are grouped into two categories: data distribution and classification results. When a collection of methods was applied to a dataset, data distribution and classification results were paired and recorded for both the proposed and the baseline methods. For the distribution, several attributes were captured, namely kurtosis, skewness, standard deviation, maximum, minimum, mean, 25th percentile, 50th percentile, and the 75th percentile. In terms of classification performance, the AUC, execution time in seconds, and the confusion matrix were recorded. A preliminary analysis was conducted to confirm a correlation between data distribution and classification results. Consequently, a heatmap was constructed based on the AUC and all the aforementioned attributes of the data distribution, as shown in Fig 3.

Fig 3 reveals a comparatively stronger association between AUC and the data distribution attributes, namely skewness and kurtosis. Interestingly, there is also a slightly weak association between the maximum value attribute and the AUC.

Furthermore, a linear regression analysis was conducted with AUC as a dependent variable and kurtosis and skewness as independent variables. The values of the coefficients for the intercept and the two independent variables, along with their corresponding t-values and p-values, were obtained. The intercept had an estimated coefficient of 0.8443, which is highly significant (p < 2e-16), indicating a strong positive relationship between the dependent variable and the constant term. The coefficient for kurtosis was −4.964e-05, with a t-value of −5.850 and a highly significant p-value of 6.93e-09, indicating that kurtosis has a negative impact on the dependent variable. The coefficient for skewness was −7.732e-05, with a t-value of −9.113 and a highly significant p-value of < 2e-16, indicating that skewness also

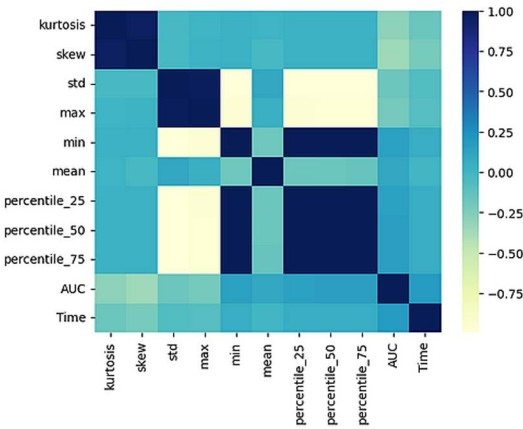

**Fig 3. Heatmap based on AUC, time, and data distribution attributes of the data distribution.**

has a negative impact on the dependent variable. Additionally, the residual standard error, a measure of the variability of the errors in the model, and the multiple R-squared values were 0.1654, indicating that the model explains only 16.54% of the variability in the dependent variable. The adjusted R-squared value was 0.1635, accounting for the number of independent variables in the model. Finally, the F-statistic and its associated p-value ( <2.2e-16) indicate that the overall model is highly significant, meaning that the independent variables as a whole are significantly related to the dependent variable.

A multiple linear regression analysis was performed by including all other variables. The intercept term was 0.8454, which is statistically significant with a very small p-value. The p-values for kurtosis and skewness remained very small, indicating that they are statistically significant predictors of AUC. However, the p-values for std, mean, and percentile_50 were relatively large, indicating that they are not statistically significant predictors of AUC at the 0.05 level of significance. The R-squared value was 0.2026, indicating that the model explains about 20% of the variability in the response variable. The adjusted R-squared value was slightly lower, suggesting that adding the additional predictors did not significantly improve the performance of model. However, the F-statistic and its associated p-value indicate that the overall model is statistically significant.

Furthermore, the classification performance of the proposed model was benchmarked against its constituent components, which comprised various combinations of rank transformation, chaining, feature extension, and feature selection, described as follows: Pre-Rank denotes the DNN model which uses input data that has initially been rank transformed followed by normalization; Post-Rank denotes the model in which normalization precedes rank transformation; "Raw and Pre-Rank" and "Raw and Post-Rank" denote the merging of features in the raw data with features generated in the former and latter methods, respectively; "Pre and Post-Rank" denotes the merging of features produced in the aforementioned ranking methods; whereas FS denotes feature selection. The results presented below are divided into two parts: the comparison of combinations in the context of single normalization methods and the utilization of chained normalization methods. This variation is meant to underscore the significance of chaining in the proposed model. The performance results of the DNN model in single normalization, Raw, Pre-Rank, Post-Rank, "Raw and Pre-Rank", "Raw and Post-Rank", and "Pre and Post-Rank" datasets are listed in Table 2. The outstanding classification results in a particular combination of methods are highlighted in bold font.

(Table 2) shows that the combination of single normalization, data ranking, and feature selection methods enhances the predictive performance of the model. While the raw data achieves an initial AUC performance of 0.800, the various combinations of single normalization methods, namely *mmn*, *vss*, *pss*, and *zsn*, with the ranking and feature selection methods have improved model performance by attaining AUC values of 0.910, 0.872, 0.871, and 0.871, respectively. However, it can also be observed that when the model is based on *mmn* only, the performance reduces to 0.757, which is about 2.4% lower than the raw data. The reduction in performance due to the *mmn* method is slightly better than that of other single normalization methods, namely *vss, pss*, and *zsn*, which attained AUC results of 0.734, 0.754, and 0.730, respectively.

It should not be surprising that *mmn*, along with the other three methods, underperforms, as research has already established that these methods are ineffective for normalizing high throughput data. These results, however, show that the combination of the four normalization methods with rank transformation, feature extension, and feature selection can substantially improve the performance of a DNN model. The observed improvements can be attributed to reduced noise in the dataset due to the rank transformation methods. Further, bolstering of

**Table 2. The AUC and time performance results of the model using raw and single normalization methods-based data and the related baseline methods.**

| | | | Raw | Pre- Rank | Post- Rank | Raw & Pre-Rank | Raw & Post Rank | Pre & Post-Rank |
|---|---|---|---|---|---|---|---|---|
| **Raw** | None | AUC | 0.800 | 0.833 | 0.834 | 0.831 | 0.838 | 0.834 |
| | | Time | 0.000 | 0.000 | 670.276 | 0.000 | 0.000 | 680.642 |
| | FS | AUC | *0.793* | 0.871 | 0.869 | 0.869 | 0.865 | **0.876** |
| | | Time | 0.010 | 0.003 | 648.421 | 0.009 | 0.005 | 605.291 |
| **mmn** | None | AUC | 0.757 | 0.795 | 0.832 | 0.758 | 0.795 | 0.797 |
| | | Time | 0.001 | 0.001 | 696.588 | 0.000 | 0.000 | 706.883 |
| | FS | AUC | 0.832 | **0.910** | 0.871 | 0.878 | 0.865 | 0.821 |
| | | Time | 0.013 | 0.012 | 708.278 | 0.005 | 0.004 | 706.883 |
| **psn** | None | AUC | 0.734 | 0.808 | 0.772 | 0.802 | 0.792 | 0.809 |
| | | Time | 0.056 | 0.056 | 722.933 | 0.000 | 0.000 | 745.229 |
| | FS | AUC | 0.853 | 0.497 | **0.871** | 0.497 | 0.860 | 0.499 |
| | | Time | 0.006 | 0.007 | 0.007 | 0.007 | 0.007 | 0.007 |
| **vss** | None | AUC | 0.754 | 0.800 | 0.820 | 0.806 | 0.821 | 0.818 |
| | | Time | 0.032 | 0.053 | 716.732 | 0.000 | 0.000 | 745.033 |
| | FS | AUC | 0.844 | 0.854 | **0.872** | 0.858 | 0.869 | 0.867 |
| | | Time | 0.006 | 0.006 | 707.504 | 0.005 | 0.014 | 718.113 |
| **zsn** | None | AUC | 0.730 | 0.816 | 0.818 | 0.790 | 0.732 | 0.823 |
| | | Time | 0.037 | 0.052 | 716.819 | 0.000 | 0.000 | 734.701 |
| | FS | AUC | 0.616 | 0.506 | **0.871** | 0.502 | 0.499 | 0.501 |
| | | Time | 0.003 | 0.003 | 0.003 | 0.008 | 0.008 | 0.008 |

feature relevance and addressing overfitting attributed to feature extension and feature selection, respectively, have also enhanced the model's performance.

It can also be observed that the combinations involving "Post-Rank" generally produce very good results most of the time. However, the combination of *mmn* with "Pre-Rank" has the overall highest performance among the single normalization methods. The authors, therefore, postulate that while the combination of methods improves model performance, there is no particular combination that will always outperform other method combinations. The reason is that the performance of the model depends on the underlying distribution of the data.

Further, while the results show that ranking of data and feature extension generally increases computation time, feature selection considerably reduces computation time, which cancels the losses due to data ranking and feature extension.

To demonstrate the effect of conventional single normalization methods on model sensitivity and specificity, the model's performance with the raw data and each conventional single normalization method was analyzed. The AUC, Sensitivity, and Specificity performance results of the DNN model with raw data and single normalization methods are shown in Fig 4.

(Fig 4) shows that all the conventional single normalization methods have not been able to improve the AUC performance of the model. Also, only half of the methods have been able to improve the model's sensitivity and specificity. While the raw data has a sensitivity score of 0.926, *vss* and *ps* have 1.000 and 0.951 scores, respectively. It can also be observed that except for the *vss* method, all conventional single normalization methods improve the model's specificity. It is also interesting to note that while *vss* has the best sensitivity value of 1.0, it also has the worst specificity value of 0.0. This case requires further investigation. However, it can be concluded that the performance of the model based on single normalization methods alone is generally not very promising.

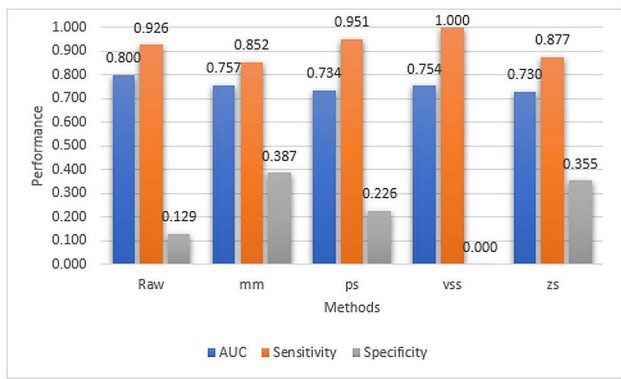

**Fig 4. AUC, Sensitivity, and Specificity performance results of the DNN model using raw data and single normalization methods.**

(Table 3) depicts the AUC and time performance results of the DNN model based on the raw data, chained normalization, Pre-Rank, Post-Rank, "Raw and Pre-Rank", "Raw and Post-Rank", and "Pre and Post-Rank". For comparison purposes, the DNN results based on the combinations that involve *mm(mmn)* are also added in the table. The outstanding classification results in the corresponding combination method are indicated in bold.

(Table 3) shows that the chaining method, as opposed to single normalization methods, in collaboration with ranking, feature extension, and feature selection, produces better classification results in the model. The highest performance in the model is based on the hybrid of the chaining method that comprises the *vss* and *psn* combination, feature extension, and feature selection, which has an AUC performance of approximately 0.923. Similarly, the chaining method involving four normalization methods, namely, *zs*, *ps*, *vss*, and *mm*, has the second-highest AUC performance of about 0.922. The results show that the combination of chaining, feature extension (by aggregating outputs of Pre- and Post-Rank methods), and feature selection produces the best prediction results in the model. However, it can also be observed that the method is computationally intensive. The combination of chaining and Pre-Rank has the second-highest classification results and a better computation time. Chaining normalization methods are associated with high performance as they can adjust the underlying data distribution by combining the strengths of the constituent methods used.

We conducted a comparative study with the raw data to demonstrate the effect of combining normalization methods, Merged rank (that is, the combination of Pre- and Post-Rank) transformation methods, and feature selection, on the AUC, Sensitivity, and Specificity of the model. The AUC, Sensitivity, and Specificity performance results of the DNN model using raw data and the aforementioned combinations are shown in (Fig 5).

(Fig 5) shows that the use of more than one normalization method through chaining, in addition to Merged rank transformation and feature selection, improves the AUC performance of the model. In particular, chaining the *vss* and *ps* methods results in the highest AUC value, approximately 0.923. The chaining of the *zs, ps, vss, and mm* methods yields the second-highest AUC value, around 0.922, followed by the *zs*, *vss*, and mm methods with an AUC of about 0.919. By comparison, applying Merged rank transformation and feature selection directly on raw data or using only the *mm* method results in lower AUC values of 0.876 and 0.910, respectively, both of which are less than those achieved with chaining. This suggests a positive correlation between the number of normalization methods used in chaining and the AUC performance of the model.

**Table 3.** The AUC and time performance results of the model using raw data in comparison with combinations of chained normalization, rank transformation, and feature selection methods.

| | | | Raw | Pre-Rank | Post-Rank | Raw and Pre-Rank | raw and Post-Rank | Pre and Post-Rank |
|---|---|---|---|---|---|---|---|---|
| **Mm** | None | AUC | 0.757 | 0.795 | 0.832 | 0.758 | 0.795 | 0.797 |
| | | Time | 0.001 | 0.001 | 696.588 | 0 | 0 | 706.883 |
| | FS | AUC | 0.832 | **0.910** | 0.871 | 0.878 | 0.865 | 0.821 |
| | | Time | 0.013 | 0.012 | 708.278 | 0.005 | 0.004 | 706.883 |
| **vss, ps** | NA | AUC | 0.8 | 0.77 | 0.83 | 0.777 | 0.831 | 0.824 |
| | | Time | 0.034 | 0.051 | 714.835 | 0 | 0 | 731.5 |
| | FS | AUC | 0.883 | 0.817 | 0.898 | 0.817 | 0.901 | **0.923** |
| | | Time | 0.006 | 0.006 | 708.393 | 0.006 | 0.007 | 720.158 |
| **zsn,vss,mm** | NA | AUC | 0.748 | 0.753 | 0.801 | 0.748 | 0.783 | 0.766 |
| | | Time | 0.001 | 0.001 | 674.009 | 0 | 0 | 703.416 |
| | FS | AUC | 0.839 | **0.919** | 0.859 | 0.88 | 0.854 | 0.856 |
| | | Time | 0.006 | 0.006 | 683.315 | 0.008 | 0.007 | 697.485 |
| **zsn,ps,vss,mm** | NA | AUC | 0.782 | 0.795 | 0.768 | 0.798 | 0.785 | 0.794 |
| | | Time | 0.001 | 0.001 | 673.6 | 0.001 | 0 | 716.644 |
| | FS | AUC | 0.845 | 0.908 | 0.827 | 0.89 | 0.81 | **0.922** |
| | | Time | 0.011 | 0.003 | 668.844 | 0.008 | 0.007 | 694.313 |

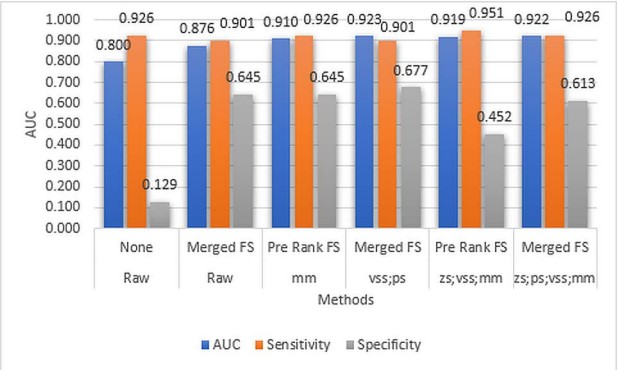

**Fig 5.** The AUC, Sensitivity, and Specificity performance results of the DNN model based on raw data and other combinations of chaining, feature selection, and rank transformation.

Furthermore, while raw data yields the lowest Specificity compared to other method combinations, Specificity still appears to depend on the number of chained normalization methods. For instance, the single *mm* normalization achieves a Specificity of 0.645, outperforming the *zs*, *vss*, and mm chain (0.452) and the *zs*, *ps*, *vss*, and *mm* chain (0.613). Similar patterns are observed in the model's Sensitivity results. Therefore, the improvements in Specificity and Sensitivity could be attributed to the combined impact of both the Merged rank transformation and feature selection methods.

Confusion matrices were also used to further analyze the performance of the model across the different permutations of the normalization methods in dataset 1. These matrices illustrate the varying degrees of performance of the classifier. For instance, matrices with higher TP and TN values and lower FP and FN values indicate better classifier performance. Conversely, higher FP and FN values and lower TP and TN values suggest poorer performance. The effect of the single normalization methods on the performance of the model is shown in (Fig 6).

From (Fig 6), it can be observed that the *vss* method has the best performance with TP, FP, TN, and FN values of 0, 0, 1, and 1, respectively. The confusion matrix shows that the method has the highest TN and lowest FN values, indicating the highest correct classifications and the lowest misclassifications. Conversely, the model based on "Raw" data has the worst performance with TP, FP, TN, and FN values of 0.07, 0.93, and 0.87, respectively, as it has the lowest TN and highest FN values, indicating a high rate of misclassification. The confusion matrices for the best performing method in the entire experiment are shown in (Fig 7).

From (Fig 7), it can be observed that the "vss, psn (M, FS)" method has the best performance with TP, FP, TN, and FN values of 0.90, 0.10, 0.68, and 0.32, respectively. The method has comparatively higher TP and TN values and comparatively lower FP and FN values. Further, although the "zsn, psn, vss, *mmn* (M, FS)" method has the highest TP (0.93), the second-lowest FP (0.07), and the lowest FN (0.39). Therefore, analysis of the confusion matrices has shown that the proposed method has outstanding performance.

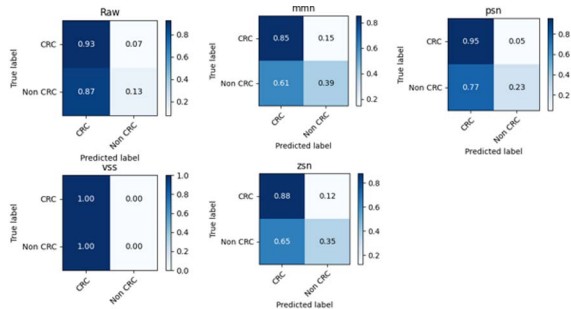

**Fig 6. A comparison of the effects of single normalization methods on the performance of the model.**

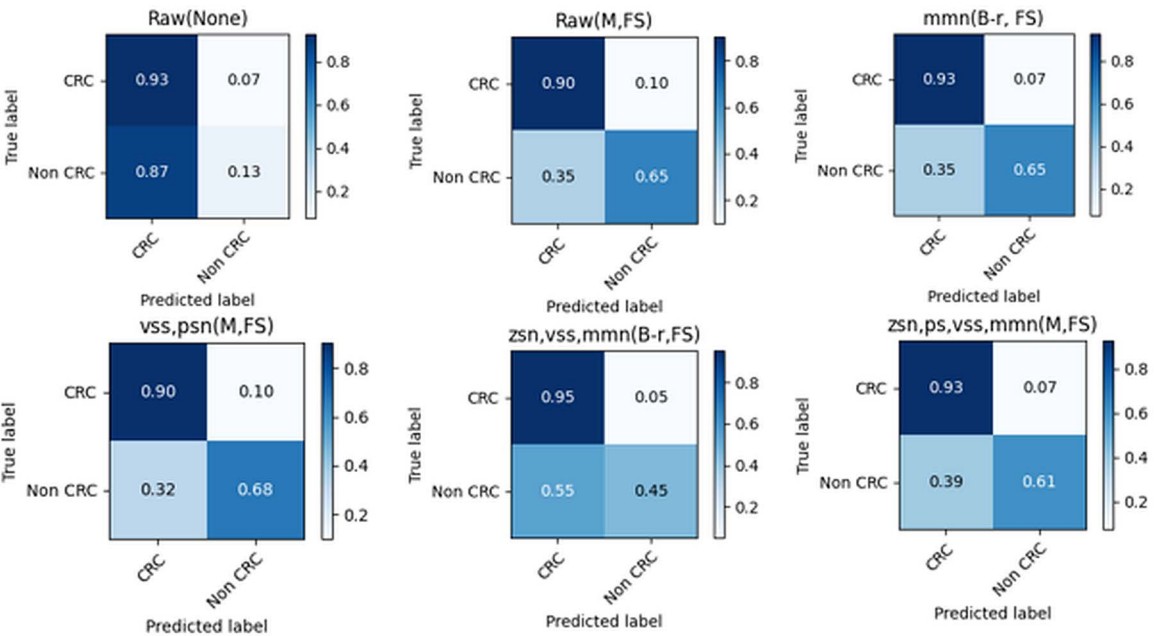

**Fig 7. A comparison of the top five performing methods in the model.**

In this work, the generalization ability of the model as a result of combining normalization methods, Merged rank transformation methods, and feature selection was also investigated using dataset 2.

The AUC performance results of the DNN model experiments with dataset 2 are shown in (Fig 8).

(Fig 8) shows that there is no single combination of methods that always outperforms all the other methods in terms of AUC. As the number of normalization methods increases, the best-performing combination keeps changing. While the gain in performance can be related to feature selection, the Merged (Pre and Post-Rank) method with the best AUC performance of about 0.975 based on the chaining of four normalization methods, namely, *ps*, *mmn*, *zsn*, and *vss*, has the overall best performance in dataset 2. Notably, the single normalization method *mmn* with the AUC values of 0.918, outperforms other single normalization methods when combined with Pre-Rank transformation and feature selection. Using the same combination, the chaining of *ps* and *mmn* methods with an AUC value of 0.929, outperforms other combinations based on two normalization methods. In the same vein, out of the three normalization methods-based chaining, the combination involving *mmn*, *vss*, and zsn has the best AUC results of 0.932. In the same context of feature selection, the post-rank method has the best performance when applied to raw data, and it has the AUC performance value of 0.891. Also, it can be observed that the Pre-Rank transformation combined with feature selection has the highest chance of outperforming other related combinations in the model. Further, just as the number of normalization methods increases in (Fig 8), so does the AUC performance of the model.

From the results, it can be observed that the combination of the Merged rank transformation methods and feature selection can be associated with the best AUC performance results of the DNN model in datasets 1 and 2. While the number and order of the individual normalization methods may vary in the proposed Merged rank transformation method, the combination generalizes well across both datasets.

## Significance of the results

The prediction results of the proposed method were subjected to significance testing to underscore its outstanding performance. In this regard, the combination of *mm* normalization and the post rank (mm-pst_r) method is used as a baseline and representative of the single normalization method due to its highest performance in the single normalization methods

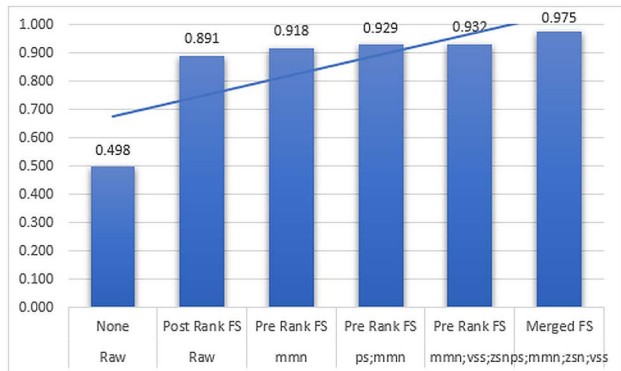

**Fig 8. The AUC performance results of the DNN model in dataset 2, based on raw data and other combinations of chaining, feature selection, and rank transformation.**

category. The selected method also outperforms the raw data. Initially, the distribution of AUC performance results for each method was examined. It was noted that certain result sets deviated from the normal distributions. As a result, the Wilcoxon ranked sum test was used to perform the significance test. This test is a non-parametric method used to test independent paired samples. The results were evaluated using a two-tailed test with a 95% confidence level. Where *pst_r*, *pre_r*, and M(merged) denote the *Post-Rank*, *Pre-Rank*, and the combination of post rank and *Pre-Rank* transformation, respectively, the findings from the significance test for the proposed method are presented in (Table 4).

(Table 4) reveals that the p-values between the *mm post-rank* (mm-pst_r) method and other single normalization techniques, namely, *mm*, *psn*, *vss*, and *zs*, are all less than 0.001. This indicates a significant performance difference between the *mm* method and both the raw data and other normalization methods. Furthermore, combinations of chained methods such as "vss,spn-M" "zs,vss,mm-pre_r," and "zs,ps,vss,mm-M" also show p-values below 0.001 when compared to mm-pst_r, signifying a significant performance improvement for the proposed method based on AUC results.

The research highlights that by integrating the strengths of chained and ranked normalization methods, alongside feature extension and selection, a robust modeling approach is achieved. Chained normalization standardizes data across various scales, enhancing consistency and aiding model convergence. Rank transformation minimizes the influence of outliers while emphasizing feature importance. Additionally, feature extension enriches the feature space, enabling the model to capture intricate relationships between variables. Feature

**Table 4. Proposed approach comprising the chaining, feature extension, and feature selection methods.**

|    | Model 1         | Model 2            | p-value |
|----|-----------------|--------------------|---------|
| 1  | Raw             | mm                 | <.001   |
| 2  | Raw             | mm-pst_r           | <.001   |
| 3  | mm              | mm-pst_r           | <.001   |
| 4  | mm              | vss,ps-M           | <.001   |
| 5  | mm              | zs,vss,mm-pre_r    | <.001   |
| 6  | mm              | zs,ps,vss,mm-M     | <.001   |
| 7  | ps              | mm-pst_r           | <.001   |
| 8  | ps              | vss,ps-M           | <.001   |
| 9  | ps              | zs,vss,mm-pre_r    | <.001   |
| 10 | ps              | zs,ps,vss,mm-M     | <.001   |
| 11 | vss             | mm-pst_r           | <.001   |
| 12 | vss             | vss,ps-M           | <.001   |
| 13 | vss             | zs,vss,mm-pst_r    | <.001   |
| 14 | vss             | zsn,ps,vss,mm-M    | <.001   |
| 15 | zs              | mmn-pst_r          | <.001   |
| 16 | zs              | vss,spn-M          | <.001   |
| 17 | zs              | zsn,vss,mm-pre_r   | <.001   |
| 18 | zs              | zsn,ps,vss,mm-M    | <.001   |
| 19 | mm-pst_r        | vss,spn-M          | 0.001   |
| 20 | mm-pst_r        | zsn,vss,mm-pre_r   | 0.008   |
| 21 | mm-pst_r        | zsn,ps,vss,mm-M    | <.001   |
| 22 | vss,sp-m        | zsn,vss,mm-pre_r   | 0.235   |
| 23 | vss,ps-m        | zsn,ps,vss,mm-M    | 0.800   |
| 24 | zs,vss,mm-pre_r | zsn,ps,vss,mm-M    | 0.155   |

selection plays a vital role in reducing dimensionality, minimizing noise, and retaining essential features given the dataset's high dimensionality.

Although the proposed method enhances model performance, noise reduction techniques pose a risk of losing crucial information. Normalization alters the data distribution, potentially diminishing the impact of outliers, while feature selection may remove features that, although less obvious, could hold valuable information. Rank transformation maintains ordinal relationships but discards actual data values, which can lead to information loss when the magnitude of differences is significant for the model. To counteract these risks, the proposed methodology employs two parallel execution paths that utilize a sequential application of normalization and rank transformation, effectively reducing noise without sacrificing essential data elements. By reversing the order of these steps in each path, two distinct feature sets are generated, aiding in the retention of vital information from the original dataset.

## 5. Conclusions

This research introduces an innovative feature engineering approach that merges two distinct feature sets from input data to construct a new dataset, addressing noise inherent in microbiome data. The methodology employs two parallel execution paths for feature transformation, applying sequential normalization methods and column-wise rank transformation. The novelty lies in the reversed execution order between chained normalization and rank transformation, resulting in unique feature sets. Feature selection is also applied to manage high dimensionality in the consolidated dataset. Experimental findings confirm that the proposed method significantly enhances the performance of Deep Neural Network (DNN) models in classifying colorectal cancer using gut microbiome data.

Despite the method's promising results, the necessity for rank transformation raises questions about its practical application in medical contexts. Assigning rank values to dataset entries depends on the existing value range in a column, complicating the ranking of unseen data. Future research will focus on strategies for generating rank values for new samples, thereby increasing the method's practical utility. Although this study makes significant progress in tackling high dimensionality and noise in microbiome data, it does not explore the identification or development of specific microbial biomarkers for colorectal cancer detection. Future investigations could consider this aspect, potentially contributing to advancements in clinical diagnosis and personalized medicine. Nonetheless, the current work emphasizes improving DL methods for disease classification, representing a significant contribution to microbiome data analysis.

## Author contributions

**Conceptualization:** Mwenge Mulenga, Arutchelvan Rajamanikam, Suresh Kumar, Saharuddin bin Muhammad, Subha Bhassu, Chandramathi Samudid, Aznul Qalid Md Sabri.

**Methodology:** Mwenge Mulenga, Manjeevan Seera.

**Writing – original draft:** Mwenge Mulenga.

**Writing – review & editing:** Mwenge Mulenga, Manjeevan Seera, Christopher Ifeanyi Eke.

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
