## [Decision Letter · Decision Letter 0]

15 Jan 2024

PONE-D-23-19630Revolutionizing Colorectal Cancer Detection: A Breakthrough in Microbiome Data AnalysisPLOS ONE

Dear Dr. Mulenga,

Thank you for submitting your manuscript to PLOS ONE. After careful consideration, we feel that it has merit but does not fully meet PLOS ONE’s publication criteria as it currently stands. Therefore, we invite you to submit a revised version of the manuscript that addresses the points raised during the review process.

The reviews for your manuscript are now in, and while some of the work has been appreciated, there are major concerns that remain to be addressed. If all of these concerns can be suitably addressed in a revised manuscript, it may be possible to consider the manuscript for further review. Sorry to be not more positive on this occasion.

We look forward to receiving your revised manuscript.

Kind regards,

Karthik Raman, Ph.D.

Academic Editor

PLOS ONE

Journal Requirements:

If you are reporting a retrospective study of medical records or archived samples, please ensure that you have discussed whether all data were fully anonymized before you accessed them and/or whether the IRB or ethics committee waived the requirement for informed consent. If patients provided informed written consent to have data from their medical records used in research, please include this information."

4. In the online submission form, you indicated that [Data can be shared upon request.].

5. Please upload a copy of Figure 3, to which you refer in your text on page 9. If the figure is no longer to be included as part of the submission please remove all reference to it within the text.

Reviewers' comments:

Reviewer's Responses to Questions

**Comments to the Author**

1. Is the manuscript technically sound, and do the data support the conclusions?

Reviewer #1: Partly

Reviewer #2: No

Reviewer #3: Yes

2. Has the statistical analysis been performed appropriately and rigorously?

Reviewer #1: Yes

Reviewer #2: No

Reviewer #3: Yes

3. Have the authors made all data underlying the findings in their manuscript fully available?

Reviewer #1: Yes

Reviewer #2: No

Reviewer #3: No

4. Is the manuscript presented in an intelligible fashion and written in standard English?

Reviewer #1: Yes

Reviewer #2: No

Reviewer #3: No

5. Review Comments to the Author

Reviewer #1: 1. Clarification of Research Objectives: It is important to further clarify the research objectives at the beginning of the paper. Make sure to elaborate on why reducing dimensions and processing microbiome data is crucial in the context of colorectal cancer (CRC). Explain in more detail why improvements in CRC classification are needed.

2. Explanation of Method Selection Process: The authors can provide a better understanding of why the hybrid feature processing technique was chosen for use in this research. Are there strong theoretical reasons or previous studies that support its use?

3. Clarification of Hybrid Feature Processing Technique: The paper includes the use of hybrid feature processing techniques. However, it would be better if the authors provide more in-depth details on how these techniques are specifically implemented. This includes what parameters are used (if any), how these parameters are chosen, and whether there are specific steps taken to optimize these techniques.

4. Deeper Interpretation: To improve this paper, a deeper interpretation of the results is needed. The authors should explain why there is a difference in performance between the various normalization methods. Are there any specific factors that influence these results? Furthermore, is there an explainable relationship between data distribution attributes such as kurtosis, skewness, and AUC?

Reviewer #2: This is the review report for the manuscript Revolutionizing Colorectal Cancer Detection: A Breakthrough in Microbiome Data Analysis submitted to PLOS ONE.

The paper highlights the transformative impact of Next Generation Sequencing (NGS) on clinical diagnosis and personalized medicine through high-throughput microbiome data. Overcoming challenges in traditional statistical and machine learning methods, the study introduces an innovative feature engineering technique that combines two sets of features, resulting in a new dataset. This approach markedly enhances the performance of Deep Neural Network (DNN) algorithms in detecting colorectal cancer using gut microbiome data, showcasing a substantial contribution to addressing the complexities of microbiome data analysis and bolstering the efficacy of deep learning methods in disease detection. There are several drawbacks of the proposed procedure. Please see the attachment for my comments to justify the usefulness of the proposed procedure:

Reviewer #3: Mulenga et al. introduced a hybrid method that integrates chained normalization, rank transformation, feature extension, and feature selection to enhance the performance of DL models in CRC classification based on microbiome data. Authors have considered 88 samples, out of which 33 were diagnosed with colorectal cancer and the remaining were non-cancer individuals. The 16S metagenome data was extracted from the sample and preprocessed through Qiime2 and R software. The preprocessed 16S metagenome data were normalized and used in the hybrid DL approach. However, there are additional matters that require clarification in an updated version. Further detailed feedback is outlined below. However, there are few areas which may be addressed by the authors for an overall improvement of their study.

1. Performance of the DNN model requires estimation of TP, TN, FP, and FN for obtaining TPR, TNR, and FPR. Require clear explanation of the DNN model based on these parameters.

2. Missing details of the gut microbial composition in CRC and healthy conditions.

3. Chain normalization was used, which includes MMN, PSN, VSS, and ZSN methods. MMADN method could improve the model's accuracy.

4. Additional analysis required for experimental validation of the DL-based colorectal cancer detection.

5. Here it was mentioned about working with a default learning rate, please add the exact value. Can also compare the training accuracy with changing this learning rate.

6. A more detailed explanation of the feature extraction stage in the DNN model would contribute significantly to the understanding of the model architecture. Could the authors elaborate on the key aspects and methodologies employed during this stage?

7. While the authors have implemented noise reduction techniques, it is pertinent to discuss the potential drawbacks of such methods. Could the authors address concerns regarding the inadvertent elimination of essential data elements during the noise reduction process and elaborate on how these concerns were mitigated in their study?

8. Add one figure showing the detailed DNN model for better understanding of the architecture.

9. May conclude with some biomarkers (Microbial abundance) for colorectal cancer detection.

10. The manuscript has several typo errors, ("Raw and pre-Ranking" and "Raw and Post-Ranking"); pre is written in lowercase, whereas the Post is in uppercase. Please maintain a consistency.

6. PLOS authors have the option to publish the peer review history of their article (what does this mean? ). If published, this will include your full peer review and any attached files.

**Do you want your identity to be public for this peer review?** For information about this choice, including consent withdrawal, please see our Privacy Policy .

Reviewer #1: No

Reviewer #2: No

Reviewer #3: No

---

## [Author Response · Author response to Decision Letter 0]

17 Oct 2024

Response to Reviewers’ Comments

(Manuscript. Ref. No.: PONE-D-23-19630)

Title: Revolutionizing colorectal cancer detection: a breakthrough in microbiome data analysis

Dear Editor:

We appreciate the time and effort exerted by the editor and referees in reviewing this manuscript. We are very thankful to the reviewers for their deep and insightful review. Our manuscript was revised to reflect their useful suggestions and comments. We hope that the paper will be considered worthy of publication as all of the comments have been accommodated. Our responses to their specific comments, suggestions and queries are presented below.

Review # 1 Comments

Comment 1.1: Clarification of Research Objectives: It is important to further clarify the research objectives at the beginning of the paper. Make sure to elaborate on why reducing dimensions and processing microbiome data is crucial in the context of colorectal cancer (CRC). Explain in more detail why improvements in CRC classification are needed.

Response1.1: We appreciate the reviewer's insightful comments and recognize the importance of clarifying the research objectives at the outset of the paper. In response, we have elaborated on the significance of reducing dimensions and processing microbiome data within the framework of colorectal cancer (CRC) detection.

Comment 1.2: Explanation of Method Selection Process: The authors can provide a better understanding of why the hybrid feature processing technique was chosen for use in this research. Are there strong theoretical reasons or previous studies that support its use?

Response1.2: We appreciate the opportunity to address your concerns regarding the method selection process, particularly regarding the rationale behind the adoption of the hybrid feature processing technique. The reason for using a hybrid method is to combine the strengths of individual methods in order to enhance model’s performance. While our method is novel we have cited appropriate literature to back up our claim as shown in line 71 to 73

Comment 1.3: Clarification of Hybrid Feature Processing Technique: The paper includes the use of hybrid feature processing techniques. However, it would be better if the authors provide more in-depth details on how these techniques are specifically implemented. This includes what parameters are used (if any), how these parameters are chosen, and whether there are specific steps taken to optimize these techniques.

Response1.3: We appreciate the reviewer's insightful comments on the need to clarificaty on the Hybrid Feature Processing Technique. We have expanded our text in line 233 to 243. that previous studies were used as the basis for identifying the optimum configuration of the proposed method. The pribious study used a brutforce method to identify optimum setups. A brutforce method is neccessitted by the fact it is currently not possible to predict the performance of a normalistion or of combintions thereof due the un predictble nature of underlying data distributions. Therefor the aim this work is demonstrate that using varius pemutations of chained normlistion methods to create additional features which are then subjected to feature selection, can significtly improve the predictive performnce of model

Comment 1.4: Deeper Interpretation: To improve this paper, a deeper interpretation of the results is needed. The authors should explain why there is a difference in performance between the various normalization methods. Are there any specific factors that influence these results? Furthermore, is there an explainable relationship between data distribution attributes such as kurtosis, skewness, and AUC?

Response1.4:Thank you for your valuable feedback. In response to the suggestion, we have revised the manuscript from line 152 to 154, line 439 to 441, and line 598 to 606

Comment 2.1: Authors needs to improved introduction to discuss the usefulness of the approach and some contemporary

work.

Response 2.1: Thank you for your valuable feedback. In response to your suggestion, we have revised the introduction to better highlight the usefulness of our approach and discuss contemporary work. Specifically, we have included additional details that underscore the challenges DL methods face when applied to microbiome data, particularly in dealing with high dimensionality and noise, which often degrade model performance. We also reference recent studies that have attempted to mitigate these issues using feature extraction techniques like autoencoders and unsupervised binning, as well as feature selection methods like filter-based approaches. Despite these advancements, we highlight that many of these existing methods address the challenges in a piecemeal manner, often failing to provide comprehensive solutions for dimensionality reduction and feature relevance. We also draw attention to baseline work proposing sequential normalization methods, which, while a step forward, do not fully resolve the issues at hand. Our study aims to build on these foundations by offering a more integrated approach to address both dimensionality and noise, contributing to the robustness of DL models in disease detection(Line 55 to 67).

Comment 2.2: The authors needs to highlight the usefulness of the approach in analyzing the high/large-dimensional micro-biome data analysis.

Response 2.2: Thank you for your valuable feedback regarding the need to emphasize the usefulness of our approach in analyzing high-dimensional microbiome data. We appreciate the opportunity to clarify this aspect in our manuscript. We have updated the manuscript in line 82, 349 to 350, 631 to indicate that by using feature selection our method addressing the issue of high data dimensionality

Comment 2.3: The paper exhibits subpar writing, characterized by the recurrence of an unsubstantiated idea lacking credible evidence.

Response 2.3: Thank you for your constructive feedback. We acknowledge the concern regarding the recurrence of an unsubstantiated idea in the manuscript. We have revised the relevant sections to provide stronger, credible evidence by inserting appropriate citations and clearer justification for the ideas presented. In doing so, we have ensured that all claims are well-supported by appropriate references and data.

Comment 2.4: The author should explicitly articulate the rationale for employing chained normalization, rank transformation, feature extension, and feature selection within the specified application context.

Response 2.4: Thank you for your thoughtful feedback. We recognize the need to clarify the rationale behind employing chained normalization, rank transformation, feature extension, and feature selection. In the revised manuscript, we have provided a detailed explanation for each of these techniques in the context of our application starting from line 590 to 596.

• Chained normalization was employed to standardize the data across different scales, ensuring consistency and improving model convergence.

• Rank transformation was introduced to reduce the impact of outliers and to focus on the relative importance of features.

• Feature extension aimed to enrich the feature space, allowing the model to capture complex relationships between variables.

• Feature selection was crucial in reducing dimensionality, minimizing noise, and retaining the most relevant features, given the high-dimensional nature of the dataset.

These techniques were chosen to optimize model performance and enhance generalization in the specified context. We hope the additional explanation strengthens the manuscript."

Comment 2.5: Crucially, the justification for employing a deep neural network is warranted in the context of a study with

a large sample size. However, the author has utilized a dataset featuring a sample size of only 512 with 324

features, which is insufficient for training a deep neural network. Author should talk about the generalization

of the approach.

Response 2.5: Thank you for highlighting the concern regarding the dataset size and the justification for employing a deep neural network. We acknowledge that deep neural networks typically perform better with larger sample sizes. However, in our study, we applied specific techniques to address this limitation, such regularization, and careful tuning of the model's complexity (Line 337 to 338). These strategies were employed to prevent overfitting and improve the generalization of the model, despite the relatively small sample size.

Comment 2.6: In the analysis of microbiome data, normalization is applied to overcome the constraint of not observing

absolute abundance data. The author should explicitly demonstrate the rationale behind employing various

normalization techniques. Currently, using two different normalization methods seems unwarranted, particularly in the context of microbiome data analysis.

Response 2.6: Thank you for your valuable feedback. We would like to clarify the rationale behind employing multiple normalization methods in our analysis. As outlined in the manuscript, we utilized a sequential arrangement of normalization techniques, where the output of one method is fed into the next. This approach was designed to address the complex characteristics of microbiome data, such as its high dimensionality and variability in distribution.

Research has demonstrated that no single normalization method consistently outperforms others across all datasets due to the diverse nature of underlying data distributions (as cited). Likewise, there is no universally optimal combination of methods for use in a chained sequence. Thus, our choice to apply multiple normalization techniques was motivated by the need to capture different aspects of the data that a single method might overlook. We have revised the manuscript to further clarify this reasoning and ensure that the rationale for using sequential normalization techniques is more explicitly justified.We appreciate your thorough review and hope that this explanation resolves your concerns.

Comment 2.7. To better facilitate the manuscript review and reproducibility of the analysis, authors should share the analysis script over any open-source platform.

Response 2.7: Thank you for your suggestion regarding the sharing of the analysis script to enhance review facilitation and reproducibility. In response, we have uploaded the analysis script to GitHub, an open-source and included the link in the revised manuscript (https://github.com/diskava/Revolutionizing-colorectal-cancer-detection-A-breakthrough-in-microbiome-data-analysis.git ). The script contains all relevant details of the data preprocessing, model training, and evaluation steps to ensure transparency and reproducibility.We hope this will assist in a more thorough review of the manuscript and ease the reproducibility of our results.

Comment 3.1: Performance of the DNN model requires estimation of TP, TN, FP, and FN for obtaining TPR, TNR, and FPR. Require clear explanation of the DNN model based on these parameters.

Response 3.1: Thank you for your valuable feedback. To respond to the comment, the confusion matrices for methods in dataset 1 have been added and analysed (line 522 to 543).

Comment 3.2: Missing details of the gut microbial composition in CRC and healthy conditions. Response 3.2: We appreciate the reviewer’s insightful suggestion regarding the inclusion of biomarkers for colorectal cancer detection. However, the primary focus of the present study is on improving deep neural network performance for classification using gut microbiome data, specifically through feature engineering and normalization methods. As such, we did not explore the identification or validation of specific biomarkers in this work. We agree that investigating microbial abundance as biomarkers for colorectal cancer is an important future direction, but it is beyond the current scope of this study. We hope this explanation is acceptable and appreciate your understanding. Also we have updated the conclusion to address this concerns 619 to 624

Comment 3.3: Chain normalization was used, which includes MMN, PSN, VSS, and ZSN methods. MMADN method could improve the model's accuracy.

Response 3.3: We appreciate the valuable feedback. As such have added text to show how MMADN was excluded from the manuscript. The facts are that in our preliminary work MMADN did not show promising results. That is why it was exclude as shown in line 222 to 226

Comment 3.4:. Additional analysis required for experimental validation of the DL-based colorectal cancer detection.

Response 3.4: We are grateful for this remark. We wish to acknowledge that Additional analysis is important for model validation. However, we wish to point that the paper uses an additional dataset besides the dataset proposed in our work. See line 184 to 185 and line 544 to 569

Comment 3.5: Here it was mentioned about working with a default learning rate, please add the exact value. Can also compare the training accuracy with changing this learning rate.

Response 3.5: We are grateful for this remark. We wish to acknowledge that there was a lapse in our earlier statement which states that the default learning rate was used when in the actual sense, the model used is based and indeed the learning rate was adopted from the experiment in the cited baseline work. As such, the text has been modified to show that that the learning rate and other parameters of the model has been adopted from previously published work. The modifications to the manuscript are in line 337 to 343

Comment 3.6: A more detailed explanation of the feature extraction stage in the DNN model would contribute significantly to the understanding of the model architecture. Could the authors elaborate on the key aspects and methodologies employed during this stage?

Response 3.6: We appreciate the valuable feedback. While the proposed method does not directly use feature extraction, we assume the reviewer meant feature selection. As such we have outlined of how L2 regularization was used in feature selection in our work ( Line 331 to 335)

Comment 3.7: While the authors have implemented noise reduction techniques, it is pertinent to discuss the potential drawbacks of such methods. Could the authors address concerns regarding the inadvertent elimination of essential data elements during the noise reduction process and elaborate on how these concerns were mitigated in their study?

Response 3.6: We appreciate the reviewer's insightful comment regarding the potential drawbacks of noise reduction techniques, particularly the inadvertent elimination of essential data elements. In our study, we were aware of this risk and took deliberate steps to mitigate it as explained in Line 597 to 605. By reversing the order of these steps in each execution path, we generated two distinct feature sets, which helped retain vital information from the original dataset. This redundancy through transformation further ensured that key data points were preserved even as noise was minimized.

Additionally, feature selection was applied to the consolidated dataset post-transformation to address high dimensionality, prioritizing features that contributed most to model performance. This selection process helped safeguard against the loss of important data elements by emphasizing those features that were most relevant to the classification task.

While we acknowledge the potential for the inadvertent removal of significant data during noise reduction, the combination of multiple transformation paths and feature selection effectively mitigated this risk in our study. We believe that these steps not only reduced noise but also preserved essential elements critical to the classification of colorectal cancer using gut microbiome data. Experimental results have demonstrated that these strategies improved model performance, further supporting the efficacy of the approach.

Comment 3.8: Add one figure showing the detailed DNN model for better understanding of the architecture.

Response 3.8: Thank you for your valuable feedback regarding the inclusion of a figure to illustrate the DNN model. We agree that a visual representation will enhance the reader's understanding of the architecture. In response, we

---

## [Decision Letter · Decision Letter 1]

11 Dec 2024

Revolutionizing colorectal cancer detection: a breakthrough in microbiome data analysis

PONE-D-23-19630R1

Dear Dr. Mulenga,

We’re pleased to inform you that your manuscript has been judged scientifically suitable for publication and will be formally accepted for publication once it meets all outstanding technical requirements.

Kind regards,

Karthik Raman, Ph.D.

Academic Editor

PLOS ONE

Additional Editor Comments (optional):

The manuscript has now been significantly strengthened, and can now be accepted for publication

Reviewers' comments:

Reviewer's Responses to Questions

**Comments to the Author**

1. If the authors have adequately addressed your comments raised in a previous round of review and you feel that this manuscript is now acceptable for publication, you may indicate that here to bypass the “Comments to the Author” section, enter your conflict of interest statement in the “Confidential to Editor” section, and submit your "Accept" recommendation.

Reviewer #3: All comments have been addressed

2. Is the manuscript technically sound, and do the data support the conclusions?

Reviewer #3: (No Response)

3. Has the statistical analysis been performed appropriately and rigorously?

Reviewer #3: (No Response)

4. Have the authors made all data underlying the findings in their manuscript fully available?

Reviewer #3: (No Response)

5. Is the manuscript presented in an intelligible fashion and written in standard English?

Reviewer #3: (No Response)

6. Review Comments to the Author

Reviewer #3: The authors have thoroughly addressed all my comments. This manuscript can now be considered for publication.

7. PLOS authors have the option to publish the peer review history of their article (what does this mean? ). If published, this will include your full peer review and any attached files.

**Do you want your identity to be public for this peer review?** For information about this choice, including consent withdrawal, please see our Privacy Policy .

Reviewer #3: No

---

## [Editor Report · Acceptance letter]

PONE-D-23-19630R1

PLOS ONE

Dear Dr. Mulenga,

I'm pleased to inform you that your manuscript has been deemed suitable for publication in PLOS ONE. Congratulations! Your manuscript is now being handed over to our production team.

Kind regards,

on behalf of

Dr. Karthik Raman

Academic Editor

PLOS ONE